# Strategic Non-Regulation as Migration Governance

**ABSTRACT**

Over the last decade, critical migration scholarship has been increasingly concerned with how state actors deploy forms of inaction and ambivalent action to govern migrants. Political scientists, IR scholars, geographers, legal scholars, sociologists and anthropologists have mobilized and developed concepts to capture such strategic non-regulation, ranging from notions of standoffishness, ignorance, indifference, ambiguity, adhocracy, and informality in political science and sociology, to necropolitics, ignorance, opacity, obfuscation, non-recording and liminality in anthropology and political geography. Scholars thus seem to agree that the strategic use of non-regulation by state actors is a significant aspect of migration governance. Yet, conceptual and methodological advances remain fragmented and scattered across geographical regions and disciplines. This paper argues that much can be gained by putting the different conceptual and methodological innovations on strategic non-regulation into dialogue. First, consolidating insights from different bodies of scholarly work moves analyses of strategic non-regulation from the fringes of migration scholarship to its center and demonstrates that strategic non-regulation is a core feature of migration governance. Second, bringing these different works together enables us to synthesize the variety of methodological strategies that scholars have devised to empirically locate the elusive phenomenon of strategic non-regulation. Overcoming disciplinary and geographical divides in the study of strategic non-regulation will also be key to advance broader social science debates on the political functionality of policy failure and on the interplay between state capacity and political will - in migration governance but also beyond.

**Keywords:**

Migration governance, inaction, ambivalence, migration, refugees

# 1. INTRODUCTION

Within migration studies, many anthropologists and socio-legal scholars have observed that migrant experiences are defined as much by the presence of regulation as by its absence or ambivalence, creating feelings of insecurity, exhaustion and disorientation (Agier, 2008; Ansems de Vries and Guild, 2019; Horst and Grabska, 2015; Tazzioli, 2019). At the same time, an abundance of studies in political science and political geography has challenged statist claims that migration is governed through coherent state action or concerted decision-making by highlighting gaps, inconsistencies, and ambivalences in migration governance (Lenner, 2019; Norman, 2020; Schiltz et al., 2018). Detailed accounts of informal governance, the consequences of piecemeal legislation, implementation gaps, and ambiguous decrees have accumulated to the extent that we can no longer see non-regulation as the exception to the rule of regulation, but rather need to acknowledge it as a core facet of migration governance.

The often implicit but quite dominant assumption in the literature has been that such non-regulation signals governance failures, deficits and limited capacity (Czaika and De Haas, 2013; Dini, 2017; Freeman, 1994; Sadiq and Tsourapas, 2021; Ulusoy, 2021) or reflects the inevitable compromises produced by political decision-making (Pugh, 2021) and the inherent complexities of modern bureaucracies (Eule et al, 2019; Gazzotti et al., 2022; Triandafyllidou, 2022). This reading of non-regulation is, importantly, often also the default understanding - or claim - of many of these governance actors themselves: 'we *want* to govern comprehensively and reliably, but we unfortunately do not have the means to do so' (see Fakhoury (2017) for illustrations) - a position projected for reasons varying from genuine conviction to attempts to maximize funds or to avoid accountability.

However, critical migration scholarship since the mid-2010s has shown that many instances of non-regulation emerge and persist precisely because they serve the interests of those seeking to control or benefit from migration (Anderson, 2014; Biehl, 2015; Chimni, 2003; De Genova, 2002; Kalir and Van Schendel, 2017). Non-regulation can have different, and mostly intertwined, functions in the governance of migration (Stel, 2020: 13-14). First, strategic non-regulation can be a form of outsourcing responsibilities to other actors, such as migration and refugee service providers or international and domestic NGOs (Davenport and Leitch, 2005; Gammeltoft-Hansen et al., 2017; Kassoti and Idriz, 2022; Norman, 2021). Second, it can be a means to maximize flexibility and leeway, where inaction and ambivalence allow actors to placate different audiences or stakeholders at the same time (Matland, 1995; Frost, forthcoming; Natter, 2021; Oomen et al., 2021). Third, it can be a tactic to avoid accountability, where vague or absent rules and mandates obstruct transparency and allow for impunity (Davitti, 2020; Costello and Mann, 2020; Feith Tan and Gammeltoft-Hansen, 2020). Fourth, strategic non-regulation can operate as a disciplinary strategy, where it creates an institutional landscape determined by uncertainty and unpredictability that increases the discretionary power of authorities and undercuts the possibilities for concerted collective action of migrant communities (Ansems de Vries and Welander, 2017; Ilcan et al., 2018; Stel, 2020).

What we call non-regulation – operationalized as state actors' use of inaction and ambivalent action – is thus not merely a ubiquitous phenomenon, it is also often strategic – something we

conceptualize as encompassing both intent, in the sense of conscious and deliberate choice, and convenience, in the sense of more systemic or structural functionality. This article thus departs from the assumption that migration is not simply governed *despite* non-regulation, but *through* it; that strategic non-regulation is a form of migration governance in its own right that deserves to become a more central and systematic object of scholarly inquiry. Yet, strategic non-regulation is tricky to empirically study. While the pertinence of strategic non-regulation resonates with many migration scholars, it is an elusive object of analysis. In fact, scholars working on different empirical contexts from across the Global South and North, especially the Middle East, North Africa and Europe, have largely stumbled on forms of strategic non-regulation without necessarily having set out to study it in the first place. To make sense of these empirical realities, they have turned to a wide variety of literatures and concepts.

Notions like standoffishness and indifference have originated in Political Science and IR, concepts such as ambiguity, adhocracy and uncertainty in Sociology and Public Administration, terms like the politics of waiting and liminality have been mobilized in Anthropology and Geography, and Law has developed concepts such as irregularity and informality. Interestingly, while both non-regulation and its strategic nature have often been cast as 'typical' for states in the Global South considered 'illiberal' and 'weak', many of the conceptualizations of strategic non-regulation in fact originate from case-studies in states in the Global North that are considered 'liberal' and 'strong' (Gatta, 2019; Kubal, 2013; Nawyn, 2016). However, these diverse scholarly works do not always engage with each other, which risks underplaying the significance of strategic non-regulation as a core facet of migration governance worldwide (Stel, 2021).

Our aim in this paper is to move beyond this nascent stage of individual and disciplinary conceptualization, to connect and synthesize these different concepts, and to offer scholars interested in studying strategic non-regulation a more concerted vocabulary and set of empirical strategies. As developed below, we propose the notion of 'strategic non-regulation' as an umbrella term to bring together a wide variety of concepts that engage with the use of inaction and ambivalent action by state actors in ways that serve these actors' stated and/or unstated interests and objectives as relating to the governance of migration.

We believe that consolidating this emerging, yet still fragmented, literature on strategic non-regulation is important for three reasons: First, strategic non-regulation has profound repercussions for migrants in that it often, although not automatically or always, diminishes their protection and undercuts their agency (Cullen Dunn, 2014; El-Sharaawi, 2015). Second, it offers new ways to escape some of the tenacious binaries defining the field of migration studies – such as the migrant/refugee, discipline/autonomy and North/South dichotomies. And third, it sheds crucial light on theoretical questions of how to consider failure and success in (migration) governance and the related intersections between capacity and will (Castles, 2017; Norman, 2021; Stel, 2020). As a result, the strategic dimensions of non-regulation have important implications for political and policy engagement that now often, and we would suggest misguidedly, considers capacity-building and funding as a panacea for 'better' migration governance, routinely avoiding the more pertinent question of political will to which we return in the conclusion.

In the remainder of this paper, we first, in section 2, provide more background on our own empirical and epistemological points of departure and on our motivation to write this paper as well as a more tangible taste of what, in our understanding, strategic non-regulation might look and feel like. In section 3, we then conceptualize strategic non-regulation as the proposed umbrella term to capture both convenient and intentional inaction and ambivalent action in migration governance. In section 4, we systematically discuss specific concepts that seek to grasp strategic non-regulation in migration governance across disciplines, identifying different approaches to conceptualizing and empirically studying 'non-regulation' on the one hand and 'strategy' on the other. While the pertinence of studying strategic non-regulation intuitively resonated with scholars and practitioners familiar with migration governance, we always met the inevitable pushback: 'but how do you *prove* it?!' We therefore conclude this section with concrete pointers on not just how to *think about* strategic non-regulation, but how to actually *empirically study* it. In the concluding section 5 we then zoom out to highlight what our analysis of this emerging body of work might mean for key debates on statehood, political will and policymaking in migration studies.

## 2.    WHY THIS PAPER: MOTIVATION AND BACKGROUND

At the time of our respective fieldworks across the 'Middle East' (Bouris et al, 2022) and North Africa over the 2010s there was no ready-made theoretical framework on strategic non-regulation within migration studies that would allow us to make sense of our respondents' statements and our observations from the field. In search of conceptual inspiration, each of us turned to a different body of scholarship outside of migration studies: to discussions around ad-hocracy in organizational and public policy studies (Natter, 2022); to research on informality and state regulation of smuggling, street vending and protests in sociology and political science (Norman, 2021); and to studies on ambiguity in political geography and ignorance studies in sociology (Stel, 2020). In the following, as an introduction to the broader aim and argument of the paper, we provide a short account of our respective intellectual journeys, specifically revisiting our own attempts to empirically 'capture' and theoretically situate strategic non-regulation - something we did not necessarily initially aim to study but which surfaced as crucial for our analyses of migration governance.

*Nora Stel*: January 2017. I sit behind my laptop, discussing with a Lebanese human rights lawyer why it seems so impossibly hard to pinpoint status regulations for Syrian refugees in Lebanon even for experienced and relevantly trained professionals who spend most of their waking hours trying to do exactly that. He suggests: "*I think again that in one way or another [uncertainty] is something that's used across our region by governments to control whether it is associations or whether it is to control migrants or whether it is people. Like, to put you in a place where, like, you don't really know; it's neither black nor white, you don't know, it's a grey area.*"

More than a decade ago I started out studying how Lebanese and Palestinian authorities interacted in informal Palestinian refugee settlements in South-Lebanon (Stel, 2015, 2016a/b, 2017). I aimed to shed light on how these different authorities engaged in terms of security and service delivery in spaces that were not officially recognized by either the Lebanese state or the United Nations and thus fell under the mandate of neither. Unsurprisingly, these interactions were

entirely informal. They were also highly unpredictable, specifically for Palestinian authorities and residents in such spaces. What stood out ten months of ethnographic fieldwork was that such unpredictability was perceived, by Palestinian interlocutors and expert observers, as not merely a contingency of informal governance, but as a more or less deliberate strategy: a way of controlling an unwanted population that was low cost and propped up the power positions of Lebanese political leaders (Stel, 2016c).

As illustrated by the citation above, this inductive finding on the significance of non-regulation in Lebanon's governance of Palestinian refugees directly resonated with the country's response to the arrival of Syrian refugees from 2011 onwards, which analysts remarked was characterized by a 'no policy policy' and 'formal informality' from the start (Nassar and Stel, 2019). Thus, while none of my initial research proposals were concerned with the phenomenon that we have opted to call strategic non-regulation here, it emerged as an empirical centrality to my work on refugee governance in Lebanon. I eventually found inspiration in the emerging scholarship on ignorance studies (Gross and McGoey, 2015; McGoey, 2019; Proctor and Schiebinger, 2008) to theorize the connections between uncertainty, ambiguity, liminality, and informality with regard to my own findings. However, while my empirical insights always intuitively resonated among the wide variety of migration scholars I presented my ongoing work to, it was hard to find a shared analytical vocabulary that could move these conversations forward. It is these experiences that motivate my contribution to this joint paper on strategic non-regulation.

*Kelsey Norman*: I began my dissertation fieldwork in Egypt in 2013 hoping to understand the prospects for refugee integration in a country without a comprehensive domestic asylum policy. Throughout the course of interviews with government officials, international organizations, local NGOs, and migrants and refugees themselves, I was repeatedly told a version of: "Egypt might not have a formal policy, but it certainly knows what's going on." A somewhat haunting anecdote that illustrates this dynamic well is from a refugee school director in Cairo, who explained to me in 2014 that his school was not registered with the Ministry of Social Solidarity, and that the Egyptian authorities had never contacted the school's administration. Nonetheless, the director received a call one morning from a polite woman at the Ministry of the Interior – speaking in perfect English – asking him to close the school that day for the safety of the students, as the government anticipated unrest from protests. The director recalled, "*I laughed, because I had actually been overseas, and I had just changed my phone number only three days earlier, but they managed to get straight to me, on my mobile*." Even though the director had never interacted with Egyptian authorities and had not properly registered the school, the authorities had clearly been monitoring the school's activities.

While the official line from state authorities is that Egypt is an overburdened country that is not capable of providing services to refugees and migrants, the anecdote demonstrates that Egypt clearly expends numerous resources to carefully observe and track the activities of the organizations that step in to provide services to migrants and refugees on the state's behalf. Instead of dismissing the Egyptian state – as well as Turkey and Morocco, the two other countries I examined in my study – as incapable of providing services or developing a clear migration and refugee policy, I chose to understand them as *strategically indifferent* toward the issue (Norman 2019, 2020). To do so, I drew on studies from political science and sociology of state inaction

(Gallien 2019), ignoring (Moss 2014; Bishara 2015) and forbearance (Holland 2017) in the realms of smuggling, responding to protests, and cracking down – or not – on street vending. Building on these concepts allowed me to conceptualize how and why states might choose to project indifference toward an issue – in this case, migration –, and what tangential benefits doing so yields for a migrant and refugee host state. My contribution to this paper stems from my motivation to use these insights to build bridges to other conceptualizations of state inaction and ambivalence.

*Katharina Natter:* In 2016, I embarked on my fieldwork to find out why there had been a liberal migration reform in authoritarian Morocco, while restrictive migration policies had remained in place in democratizing Tunisia. In the many hours of interviews with civil society and political actors, one rather unexpected insight stood out: regardless of the political regime in place (a monarchy or a democratic government) and the broader policy goal pursued (migration restriction or liberalization), authorities preferred to govern migration through decrees, temporary exemptions and informal arrangements – rather than through law-making (Natter, 2021). Talking about the expansion of migrants' rights in Morocco, one respondent criticized: *"They occurred outside of the law, there is no legal basis for them. That's the fragility of it – it can be gone as quickly as it came."* And in Tunisia, one respondent concluded: "*Tunisia does not want to be held accountable by something that is written, that is palpable, like a residence card, a law, a circular [...] Whatever domain you are looking at, you will find the same logic, keeping the ambiguity, so that discretion remains the basic framework for managing migration*."

While I could relate these insights to scholarly discussions on discretion and ambiguity in migration policy *implementation*, none of these terms sufficiently captured the strategic, intentional dimension of what happened at the level of *policy-making*. I therefore turned to the literature on ad-hocracy in organizational studies and public policy (Cullen Dunn 2012; Miller 1986; Rourke and Schulman 1989). Although not specific to migration, it provided me with conceptual tools to make sense of the intentionality with which policy actors in Morocco and Tunisia used executive politics, exemption regimes and case-by-case arrangements to strengthen the state's margin of manoeuvre over immigration and to navigate external and bottom-up pressures for more immigrants' rights by performing yet ultimately avoiding compliance (Natter 2021). With migration policy scholarship expanding rapidly over the past years, I saw more and more parallels between my findings on Morocco and Tunisia and policy dynamics in other countries, be they in the Global North and South. However, a more concerted dialogue between studies did not emerge because researchers used very different concepts to make sense of their findings. As Nora and Kelsey, I grew increasingly frustrated of the limitations of not having a common vocabulary. I therefore hope that our paper will provide a conceptual roadmap for the burgeoning migration policy literature to more systematically study, contrast and compare, as well as theorize what we call here strategic non-regulation.

## 3.    THE UMBRELLA CONCEPT: STRATEGIC NON-REGULATION

We have started off this paper noting that the salience of strategic non-regulation is evidenced by the fact that it comes up, albeit under different terms, across disciplines and geographies. We find instances of strategic inaction and ambivalence in different disciplinary contributions to migration studies and in cases across the Global North and South. This also means that our discussion below encompasses a vast array of different practices and institutions. We are convinced that it is analytically productive to consider them under one and the same umbrella term - strategic non-regulation - because, as argued above, it allows us to see the significance as well as the variation of strategic non-regulation and because it makes possible a methodological cross-fertilization in terms of rendering strategic non-regulation empirically researchable. In this section we motivate our choice for the term strategic non-regulation in migration governance as an umbrella concept and operationalize its two core components – non-regulation and strategy – to provide the foundation for the remainder of this paper.

### *Migration governance*

To start off with an important disclaimer: while part of our own work has predominantly focused on refugees as a (supposedly) distinct category of migrants, we do not limit our discussion to forced migration as strategic non-regulation is a constant in the governance of people on the move more broadly (Agier, 2011; Dahinden et al., 2019; Robertson, 2019). Nor do we think strategic non-regulation is unique to the field of migration. While the phenomenon of migration combines certain aspects that make it perhaps more prone to non-regulation (volatility, transnationalism, 'crisis'), we see non-regulation as relevant to all domains of governance – whether it is people, capital, labor, or territory. While we focus our analysis on the governance of migration, therefore, we are convinced this focus produces insights that resonate beyond this field, as we reflect on in the conclusion.

We consciously use the term 'governance' to signal that regulation, in this case of the field of migration, is not solely the domain of 'government' (i.e. the state) but rather that it emerges through the interaction of a complex assemblage of public, private, and societal stakeholders ranging from politicians, bureaucrats, NGOs and CSOs, humanitarian agencies, entrepreneurs and businesses, and international organizations. Yet, we also consider that within these assemblages, state actors take on a particularly central role in migration governance. Indeed, the very phenomenon of migration only exists due to the prevalence of an international nation-state system and its bordering practices (Van Houtum and Van Naerssen, 2002; Van Houtum and Bueno Lacy, 2020). Despite developments of governance moving 'up' and 'down' from the national 'level' and the increasingly transnational nature of migration governance, it is arguably still state actors that shape the parameters for migration governance (Betts, 2011; Hansen and Stepputat, 2005). In this paper, we therefore focus on non-regulation by state actors.

### *Non-regulation: inaction and ambivalence*

Non-regulation is per definition elusive as it regards things that are absent, partial, and vague. To clarify the scope of our discussion, we follow Stel (2020) in conceptualizing non-regulation as combining inaction and ambivalent action. Inaction refers to the absence of decisions and actions (Barber, 2017) and 'not dealing with' modalities of governance (Kalir and Van Schendel, 2017: 6);

to instances where state actors have the mandate to act but do not do so (McConnell and 't Hart, 2014) and where they engage in nonperformativity (Ahmed, 2006 in Stel, 2020), claiming to act while remaining inactive. Ambivalent action concerns the inconsistent or ambiguous nature of decisions and actions (Best, 2012; Norman, 2017; Pinker and Harvey, 2015), where decisions are conditional or temporary, regulations are vaguely formulated, mandates left imprecise, and implementation guidelines are contradictory or partial (Nassar and Stel, 2019). Crucially, these two dimensions of non-regulation often go hand in hand. Inertia and avoidance are never total and often enable forms of ambivalence. Our understanding of non-regulation encompasses both inaction and ambivalence in formal law and policy-on-paper, as well as in informal policy and practice. As the discussion later on will show, formality and informality are tightly related to one another: inaction in the formal realm might generate ambivalence in both formal and informal dimensions of governance; ambivalence in formal laws and policies might legitimize or incentivize inaction in formal governance and further ambivalence in informal practices.

### *Strategy: intent and convenience*

Inaction and ambivalence – in migration governance as in other realms of governance – is neither surprising nor necessarily problematic. As we flagged in the introduction, non-regulation is a natural contingency of political consultation and compromise, of the temporal divergencies between decision-making and implementation, of the administrative complexities of modern multi-scalar bureaucracies, and of the inherent scarcity of resources. Here, however, we are interested in *strategic* non-regulation. This excludes the extensive scholarship that considers forms of non-regulation in migration scholarship solely or primarily from the angles of complexity, capacity, and contingency. Instead, we discuss and synthesize scholarship that has acknowledged that many of the gaps and loose ends we consider as forms of non-regulation *are* contingent on finite resources and inevitable organizational intricacies and path-dependencies, but shows us that this only tells part of the story, that many forms of non-regulation are at least partially either created or maintained because they serve the stated or unstated aims of government actors better than regulation – even if strategic non-regulation may also be beneficial to migrants and can thus be benign rather than disciplinary, something we return to in the concluding section of the paper.

Strategy, in this paper, in a basic sense refers to the productive and functional nature of non-regulation, with various forms of inaction and ambivalence as discussed in the next section potentially serving as tactics towards these strategies. As we noted above, non-regulation serves interests: it allows for de facto outsourcing, for flexibility, for avoiding responsibility and accountability, and for controlling migrants and other stakeholders in migration governance. Calling non-regulation strategic thus aims at revealing and tracing these functions. All the concepts we discuss in section 4 under the umbrella term of strategic non-regulation assume, in essence, that state actors benefit from non-regulation and thus might seek to create and extend it, albeit in different ways and on different levels. In this sense, strategy is often intuitively associated with intent, by which we mean the deliberate, conscious choice for a particular course of action or inaction, clarity or ambivalence (Natter, 2021), or with the purposeful pursuit of explicit objectives. This is certainly a crucial aspect of our understanding of strategy.

Yet we complement this intentional aspect of strategy with an additional understanding of strategy as convenience (Stel, 2020). From this perspective, non-regulation is understood as having functions and serving interests, but the focus is less on tracing the direct agency behind these interests and functions and linking them to specific state actors and more on understanding the systemic dimensions of forms on inaction and ambivalence. Strategy as convenience seeks to understand how non-regulation follows from, but also upholds, legitimizes, and reproduces, migration regimes that serve the interests of state actors over those of migrants (Lemaire et al., 2021; Tazzioli, 2019).

These two forms of strategy complement each other in terms of the units of analysis and data sources they privilege and their understanding of the types and extent of 'proof' or 'evidence' for identifying 'strategy.' As such, considering them jointly allows for a more comprehensive understanding of both the implicit and explicit, the agential as well as structural, forms of strategy. Indeed, as our discussion of the literature in section 4 demonstrates, it is precisely in reading together these different understandings of strategic non-regulation that we can bridge disciplinary divides, connect fragmented discussions, and reveal the centrality of strategic non-regulation to migration governance at large.

### *Policy-making and policy implementation*

Finally, we distinguish between forms of strategic non-regulation that focus primarily on its operationalization at the level of policymaking versus policy implementation (Schulz, 2020). Policymaking includes the construction of laws, policies, decrees, regulations, and other forms of governance, while policy implementation examines the enactment, carrying out and completion of such decisions, whether through the actions of bureaucrats, state security agents, asylum officers, or the ways that such policies are experienced by individual migrants (Dekker, 2017). Admittedly, this can be a difficult delineation to make, and we acknowledge that some concepts focus on both policymaking and policy implementation. Nonetheless, including this distinction helps us to better understand how scholars choosing to focus primarily on either policymaking versus implementation develop a concept, marshal evidence in support of it, and examine the effects on various state and migrant actors.

## 4.      THE STATE OF THE ART OF STUDYING STRATEGIC NON-REGULATION

Scholars have explored strategic non-regulation through different lenses and from different disciplinary and methodological angles. In this section, we put into dialogue relevant work on strategic non-regulation in migration governance by asking two fundamental questions that follow from our theorization above: How do relevant concepts define and demarcate non-regulation in relation to inaction and ambivalence? And how do relevant concepts empirically operationalize and capture the strategic nature of non-regulation in relation to intent and convenience?

This exercise merits some preliminary reflections and disclaimers. First, we do not see the below exploration as a judgment of what is the right or best way of studying strategic non-regulation.

Rather, we start from the assumption that the different disciplinary and epistemological positions underlying the various concepts we discuss – with their differing levels of analysis, data sources, case material and theoretical prioritizations – each bring in essential pieces of the puzzle that strategic non-regulation constitutes. This also means that we do not aim or claim to provide a comprehensive overview of all conceptualizations potentially falling under our umbrella concept. Rather, we have pragmatically selected various conceptualizations that we think illustrate specific dimensions of strategic non-regulation particularly well.

Second, the grouping of concepts in this section has been developed inductively and is not an attempt at providing a typology of strategic non-regulation. We acknowledge that many of the conceptualizations of non-regulation that we discuss address both inaction and ambivalence, and consider strategy as partially intentional and partially systemic. We use intent and convenience here as organizing principles to indicate more "input" oriented understandings of strategy that focus on the origins of non-regulation, and more "output" oriented understandings that engage with the consequences of non-regulation. We do this not to argue that these are opposing schools of thought or that they are mutually exclusive phenomena. In fact, almost all the concepts we discuss here under the umbrella term of strategic non-regulation assume, in essence, that state actors might benefit from non-regulation in some situations, and in different ways and on different levels could seek to create and extend it. Where they differ – and this is predominantly an epistemological concern – is to what extent this strategic "will" can be "proven." Some approaches are simply not concerned with substantiating evidence on the intent and will of state actors to enact a non-regulatory policy even if they might be convinced that such will is present, but instead focus mostly on its effects. In other words, they deduce the convenience of non-regulation for authorities from the impact of non-regulation on migrants and refugees. That we nevertheless categorize various concepts – or rather groups of concepts – based on the emphasis they put on either one of these dimensions is because we are convinced that this can help us to theorize, operationalize, and empirically identify different dimensions of strategic non-regulation.

Third, it might be helpful to further delineate our umbrella concept of strategic non-regulation by further illustrating how not all concepts relating to the state's partial or non-presence in the field of migration governance fits under this umbrella term. For example, Pugh (2018) uses the term "invisibility bargain" to describe how Colombian migrants and citizens in Ecuador live with an unwritten set of expectations in which the host state and society implicitly accept the presence of foreign migrants, as long as these migrants are seen to bring economic benefits to the country and maintain political and social invisibility. Colombian migrants also employ strategies to reduce social distance, minimize differences, and build relationships and coalitions that allow them to negotiate informally with non-state allies and intermediaries in the host society. While part of what allows the invisibility bargain to function is the absence of formal state policies, the concept of invisibility bargain is more concerned with and reliant upon societal reaction, therefore not fitting within our conception of strategic non-regulation.

Ultimately, this section aims to structure the discussion of commonalities, respective strengths and distinctions between the different concepts falling under the umbrella term of strategic non-regulation in order to initiate a common conversation across disciplinary and geographic divides. For each dimension of strategic non-regulation, we discuss two key concepts we consider

particularly helpful in analytically operationalizing and empirically identifying inaction or ambivalence as well as intent or convenience. Thus, rather than trying to exhaustively *represent* all instances of strategic non-regulation, the sixteen examples we discuss *illustrate* how scholars operationalize and empirically capture the eights dimensions of strategic non-regulation. These dimensions complement each other and conceptually overlap, but, we argue, distinguishing between them is useful to peer inside the "black box" we face when studying a mode of governance that is elusive – and, crucially, in many cases meant to be elusive.

| | Inaction | | Ambivalence | |
|---|---|---|---|---|
| | *Policymaking* | *Policy implementation* | *Policymaking* | *Policy implementation* |
| **Intentional** | **Strategic indifference**<br><br>**Standoffish policymaking** | **Non-recording**<br><br>**Strategic ignorance** | **Ad-hocracy**<br><br>**Calculated informality** | **Politics of discretion**<br><br>**Governance by arbitrariness** |
| **Convenient** | **Necropolitics**<br><br>**Politics of uncertainty** | **Strategic ignorance**<br><br>**Politics of non-knowledge** | **Semi-legality**<br><br>**Strategic institutional ambiguity** | **Protracted uncertainty**<br><br>**Politics of disorientation** |

Table 1: Strategic Non-Regulation and its Dimensions

### *Inaction*

As conceptualized above, inaction is one of the two key dimensions of strategic non-regulation. In this vein, we identified four sets of concepts that are particularly helpful in revealing the intentional and convenient aspects of such inaction, be it at the level of policymaking or implementation.

#### *(1) Intentional inaction in policymaking*

We begin by examining concepts that address manifestations of intentional inaction in the realm of policymaking, or the 'input' side of strategic non-regulation (Boswell, 2007; Boswell et al, 2011). Two concepts drawn from the discipline of political science and both looking at the Middle East and North Africa use similar approaches to address national-level inaction with regard to migrant and refugee policymaking. Specifically, we highlight "strategic indifference" as developed by Norman (2019, 2020) and "standoffish policy-making" from Mourad (2017). Norman and Mourad explicitly address the question of intent in inaction by showing how the absence of action or intervention in the field of migration governance is not the result of lacking capacity, but beneficial for state actors in two ways: by lowering the level of resources required to keep control over migration and by minimizing the responsibilities for governing it. Concretely, they point at changes in state policy that move away from non-regulation to argue that inaction was a choice rather than an indication of non-capacity

In pinpointing inaction, Norman (2019, 2020) engages with the experiences and motivations of policy-makers in Egypt, Morocco and Turkey to explain how state actors proclaim that they are "strategically indifferent" to the presence of migrant and refugee groups, thereby inviting IOs and NGOs to step in and provide basic services. To operationalize intent, Norman argues that in each of her three cases, changing from a policy of strategic indifference to other more resource-intensive policies (a liberal policy or a repressive policy) demonstrates that indifference was a choice. She also argues that state actors' willingness to use resources to monitor migrants, refugees and NGOs in contrast to its unwillingness to expend resources on service provision for these groups shows that they are intentionally exercising restraint rather than being unable to act.

To empirically locate inaction, Mourad (2017) builds on work on 'no-policy-policies' (El Mufti, 2014; Ghaddar, 2017). She introduces the concept of "standoffishness" to capture how, in the early period of Syrian arrivals, Lebanese central authorities preferred to have minimal involvement in the regulation of Syrians within their borders and thus abstained from policy-making in this realm, enabling – and at times encouraging – this space to be taken up by local and international authorities. getting at intent, similar to Norman, Mourad points to Lebanon's ability to move toward a more actionable and direct policy in 2014 in regard to Syrian refugees demonstrates that the state's previous inaction was not due to incapacity but rather intentional restraint.

### *(2) Intentional inaction in policy implementation*

A similar set of concepts examines intentional inaction but primarily focuses on policy implementation (outputs) rather than policymaking (inputs). Although coming at it from different disciplinary perspectives, conceptualizations of "non-recording" (Rozakou 2017), "irregularity as statecraft" (Kalir and Van Schendel 2017), "obfuscation" (Tazzioli 2020) and "strategic ignorance" (Scheel and Ustek-Spilda 2019) all focus on the (non-)creation and suppression of data as part of a wider politics of (non)-knowledge. They evidence how knowledge is intentionally avoided, suppressed, ignored or silenced by state actors tasked with implementing migration policies to increase their legitimacy in inter-actor dynamics or to avoid responsibility and accountability. Here we specifically discuss the notion of "non-recording" and "strategic ignorance" as developed by Rozakou (2017) and Scheel and Ustek-Spilda (2019), respectively, to tease out specific analytical strategies to study intentional inaction in migration policy implementation.

Rouzakou (2017) introduces the concept of "non-recording" to capture the practices used by Greek officials toward asylum seekers as a form of inaction in the realm of policy implementation: while there is a policy directive to register, this is systematically avoided in practice. Non-recording allowed the Greek state to avoid responsibility and generate leverage vis-a-vis the EU, while simultaneously allowing migrants free mobility. To tackle intent, Rozakou (2017: 37) considers that non-recording a form of statecraft rather than state failure, something she substantiates through ethnographic fieldwork and formal interviews with agents of the state who criticized what they considered to be the government's intentional strategy of non-recording and who found "the practices of the state (which they embodied) illegible…[and] also totally 'irregular'".

In an alternative approach to intentional inaction in migration policy implementation, Scheel and Ustek-Spilda (2019) leverage ignorance studies. They understand "ignorance" in migration

governance to be a particular type of non-knowledge that is actively produced and which involves the obfuscation or suppression of otherwise available knowledge. They pinpoint such strategic ignorance through identifying four different ways of perpetuating non-knowledge, the intentionality of which is approached towards assessing the degree of consciousness with which this is done: (1) omitting the significant gap between recorded immigration and emigration events, (2) compressing different accounts of migration into one "world migration map," (3) deflecting knowledge about the specificity of different methods to production sites of statistical data, and (4) using metadata for sanitizing the statistical production process of any messy aspects. To evidence those mechanisms, they conducted a multi-sited, collaborative ethnography and studied the practices of statisticians, data scientists and other stakeholders through interviews, participants observations and workshops, as well as an analysis of produced documents, showing how numerification is performative and how strategic ignorance can result from the "non-transfer" of knowledge from one epistemic community to another (Scheel and Ustek-Spilda, 2019: 668).

### (3) Convenient inaction in policymaking

Instead of focusing on intention, other scholars have approached inaction as a form of strategic non-regulation through the lens of convenience, exploring the more systemically functional aspects of inaction through the notions of the "politics of uncertainty" (Stel 2020) as well as "necropolitics" and the "politics of abandonment" (Estevez, 2021; Pinelli, 2018; Round & Kuznetsova, 2017; Davies, Isakjee, and Dhesi, 2017). To identify helpful ways to study such convenient inaction at the level of policy-making, we offer a closer look at the concepts of "necropolitics" as used by Davies, Isakjee, and Dhesi (2017) and the "politics of uncertainty" as developed by Stel (2020).

Bringing Mbembe's (2003) notion of necropolitics – the deliberate 'letting die' (refusing to save, rather than active killing) of populations – to the field of migration studies, Davies et al. (2017) see inaction as a form of structural violence, looking both at state withdrawal as well as state action. The authors see such abandonment and withdrawal as deliberate, but they are analytically interested in evidencing the convenience of violent inaction rather than in proving intentional design. Specifically, Davies et al. (2017) study such convenience by tracing 'the connections between the political abandonment of refugees and the physiological violence they suffer:' showing that migrants suffer not as a consequence of mobility, but as a consequence of state practice. They study such convenience through two steps. First, they show that the violence generated by abandonment serves authorities' explicated aim of 'coercing onward migration.' Davies et al. (2017) conclude that the structural abandonment of migrants they identify can only be understood in relation to this explicit aim of discouragement and expulsion. Second, they demonstrate that authorities are exercising willful ignorance toward the violence generated by abandonment, 'turning a blind eye' to conditions known and publicly documented by other relevant actors.

Stel (2020) uses the notion of "the politics of uncertainty" to argue that while the precarity and uncertainty that governs the lives of Palestinian and Syrian refugees in Lebanon is typically blamed on a lack of state capacity and political and economic "crises," these explanations mask the political utility of refugees' uncertainty for state actors. While the notion of a "politics of

uncertainty" encompasses inaction and ambivalence, intent as well as convenience, and policy-making in addition to policy implementation, we here leverage it to illustrate fruitful approaches to studying convenient inaction in policy-making. Stel offers us a framework to home in on inaction in policy-making by assessing the presence or absence of laws and decrees in the domains of refugee status (whether there are legal guidelines for residency and asylum), refugee shelter (how encampment or self-settlement is officially regulated), and refugee representation (who is formally recognized as representing refugees in interactions with the state). To determine the convenience of such state inaction in terms of refugee status, shelter, and representation, Stel proposes to turn to ethnographic work on refugees lived experience and interviews with refugee representatives and 'experts' to look at the way in which these forms of inaction contribute to controlling, exploiting, and/or expelling refugees; tracing interests to demonstrate strategy.

### (4) Convenient inaction in policy implementation

A growing literature concerned with unknowing, non-knowledge and ignorance in migration studies has tried to get at inaction at the level of policy implementation, sometimes seeking to show intent and at other times aiming to demonstrate convenience (Bradley, 2023; Eule et al. 2019, Canning, 2018; Krause, 2022; Scheel, 2021). To offer specific ways to operationalize inaction and pinpoint its convenient aspects in policy implementation, we here specifically highlight recent work on the "politics of non-knowledge" by Aradau and Perret (2022) and "strategic ignorance" by Borrelli (2018).

Aradau and Perret (2022) present the production, contestation and circulation of what is not known or claimed to be unknown as an entry point for studying inaction in the sense of absence. Their empirical focus is on what they call border controversies, or disputes and disagreements about knowledge claims regarding specific migration governance practices, and they focus specifically on migrants' status determination processes and court cases on subsidiary protection. While the absence of knowledge among authorities is construed as 'error,' mistakes made in good faith, migrants who lack knowledge are considered to have enacted fraud in bad faith. These epistemological differentiations (re-)produce normative and legal hierarchies and power relations: where errors are correctible and thereby assume and produce credibility, 'fakes' assume deception and undermine credibility. Tracing the processes through which absence of knowledge is designated as either 'error' or 'fake' can hence serve as an analytical instrument to empirically reveal the different forms in which unknowing as a form of inaction is convenient in dynamics of migration policy implementation.

Borrelli (2018) draws on ignorance studies to show how we might account for the strategic aspects of inaction at the policy implementation level. However, where we brought in Scheel and Ustek-Spilda (2019) to help us concretize the potential intentionality behind inaction in policy implementation, we turn to Borrelli's work on strategic ignorance to tease out the ways in which the strategic nature of ignorance as inaction can be studied as a form of convenience as well. Based on work in Switzerland, Latvia and Sweden, Borrelli (2018) views "strategic ignorance" as a tool used both consciously and unconsciously by state officials to manage the moral and emotional implications of their work, allowing them to deal with difficult, sensitive, or uneasy tasks, while simultaneously subjecting migrants to precarious legal statuses and structural violence.

Strategic ignorance, here, is a coping mechanism more than a disciplinary strategy. Yet, she points out, its effects serve functions of strategic non-regulation such as outsourcing, creating leeway, avoiding responsibility, and undermining migrants' collective action that are more than byproducts of bureaucrats' struggles. Borrelli's reading of ignorance as convenient thus also points to the structural incentives that uphold ignorance and prop up the convenient status quo of a repressive migration regime, but cannot be traced back to a single identifiable actor.

### *Ambivalence*

In addition to inaction, we have identified ambivalence as the second crucial pillar of strategic non-regulation. Here too, we discuss four sets of concepts that conceptualize the intentional and the convenient dimensions of strategic ambivalence and engage with its dynamics on either a policy-making or implementation level.

#### *(5) Intentional ambivalence in policy-making*

In investigating the intentional aspects of ambivalence as manifest in decision-making processes, scholars have advanced notions such as "ad-hocracy" (Carpi, 2015; Natter 2021), "calculated informality" (Mielke 2022), "strategic institutional ambiguity" (Tekin 2022; Frost forthcoming; Memisoglu and Ilgit, 2017) or "uncertainty" (Volinz 2021) to bridge the domestic and international aspects of migration governance. These conceptualizations explicitly discuss how ambivalence is deliberately used by state actors across various phases of the policy-making process to navigate domestic and international policy audiences. As such, these complement how Norman and Mourad have pinpointed intentional non-regulation in the form of inaction. In order to shed more light on ways in which to empirically locate and analytically process forms of intentional ambivalence in policy making, we discuss the notions of 'ad-hocracy' and 'calculated informality' in more detail.

Natter (2021) locates ambivalence in the manner in which authorities use temporary and conditional and exceptional policy formats. She evidences how state actors secure their power over immigration by deliberately preferring "ad-hoc" policy tools that increase their governance leeway by not setting anything in stone. State authorities deliberately avoid parliamentary law-making, and instead deliberately mobilize policy tools such as (1) executive politics and so-called rule by decree, (2) exemption regimes, and (3) case-by-case arrangements. These 'ad-hoc' policy tools, Natter shows through careful document analysis and interviews, are consciously – i.e. intentionally – selected to allow Moroccan and Tunisian officials to selectively respond to external and bottom-up demands for more immigrant rights while at the same time securing the state's margin of maneuver over immigration.

Mielke (2022) engages with similar dynamics in national and international policymaking, but opts for a different vocabulary to reveal ambivalence and interrogate its potential intentionality. She discusses "calculated informality" in the context of Pakistan's rhetoric of returning Afghan nationals versus its de facto policy of non-return. Ambivalence, then, is located in the gap between public rhetoric and formal policy. Calculated informality entails state actors' successful navigating of the domestic and geopolitical arena over time based on practices of deregulation, opacity and ambiguity, constituting a strategically and purposefully applied governance mechanism reflecting

a state-sanctioned mode of deregulation. Mielke opts to get at intentionality by reconstructing the various policy options Pakistan has had since 1980 in regard to the return of Afghan nationals and assessing the extent to which eventual choices followed from conscious (informal) decision-making or structural path-dependency. Identifying such 'critical junctures' in policy-making, she demonstrates how the state has utilized informality and the de facto non-return of Afghans as an instrument that allowed the Pakistani state to appease both domestic political audiences demanding return and international audiences opposed to return.

(6) *Intentional ambivalence in policy implementation*

The intentional aspects of ambivalence in migration governance are also evident in policy implementation. The notions of "politics of discretion" and "arbitrariness" have become commonplace in a broader literature engaging with discretionary power in migration policy implementation (see for instance Alpes and Spire (2014), Darling (2022), Heyer (2022), Oomen et al. (2021), Darling 2022, Mindus 2020, Schultz (2020); McClusky, 2020; Pannia, 2020; Rigo, 2020). These works focus on the dynamics of street-level bureaucracy, dissecting the gap between policies-on-paper and implementation practices, as well as the link between formal laws and informal practices that characterize the so-called "grey area of government". In contrast to other groups of concepts, these works of anthropology and socio-legal studies encompass both studies of EU and non-EU countries, using similar vocabularies and methodological approaches. Here, we consider work on "politics of discretion" as discussed by Oomen et. al (2021) and on "governance by arbitrariness" as proposed by Mindus (2020) as exemplary for this broader group of concepts around intentional ambivalence at the implementation level.

Mindus (2020) approaches ambivalence as manifested in legal arbitrariness – characterized by a 'lack of reason-giving, legitimacy, well-foundedness' and associated with unpredictability and unboundedness. This allows her to distinguish between discretion and arbitrariness (with the former entailing a legally defined and demarcated range of flexibility for decision-making grounded in competence and proportionality). Legal arbitrariness locates ambivalence in the breaches of legality (as being at odds with a defined legal rule, often at a superior level), rationality (as evident in internal inconsistencies between declared means and ends), and egality (as seen in discrimination either within or before the law). The intentionality of legal arbitrariness is considered by Mielke as process-based (generic disregard for the law) and interest-based (disregard for the law with regard to context-specific political objectives). She evidences how legal arbitrariness has been deliberately produced and used 'by states in order to obtain a variety of border control effects,' and approximates intent by asking whose purposes are leading in the production of certain laws and by juxtaposing the extent to which laws appear inconsistent for those subjected to them and for those producing them.

Coming at it from a socio-legal perspective, Oomen et al (2021) use the politics of discretion to study ambivalence by looking at the strategies of "divergence" from national-level migration policies used by local authorities in Greece. Distinguishing between explicit and implicit divergence, as well as divergence within or outside the law, they identify four main strategies of ambivalence in implementation: (1) "defiance", in which local authorities vocally oppose national policies (explicit and extra-legal), (2) "dodging", whereby local authorities challenge national

policies by evading attempts of the central government to exercise their authority (implicit and extra-legal), (3) "deviation", whereby local authorities maximize their legally defined space of discretion (explicit and within the law), as well as (4) "dilution", whereby local authorities deliberately diverge from national policies but without challenge any existing norms (implicit and within the law). By tracing the decision-making processes through which local authorities navigate and diverge from central-level policies and norms, Oomen et al. (2021) evidence the intentional aspects of policy ambiguity.

### (7) Convenient ambivalence in policy-making

In contrast to works pinpointing intention, another set of concepts has approached the strategic element of ambivalent action through convenience. Here legal (anthropology) scholars' work on "liminal legality" (Menjivar 2006), "legal illegality" (Rigo 2011) or "informal legal orders" (Urinboyev 2020; Üstübici, 2019), geographers theorization of spatial liminality (Katz, 2019; Papoutsi et al., 2019; Oesch, 2017; Ramadan and Fregonese, 2017; Sanyal, 2018) and work on institutional ambiguity (Martin 2015; Oesch 2017; Stel 2016) have been of great inspiration. Here we discuss Kubal's (2013) notion of "semi-legality" and Nassar and Stel's (2019) interpretation of "strategic institutional ambiguity" to tease out specific analytical strategies to study convenient ambivalence in migration policy making.

The idea of semi-legality offers a legal perspective on ambivalence that locates such ambivalence in the absence of binary il/legality. It points at the relevance of not discarding legal ambiguities and 'in-between statuses' as part of a bulk notion of illegality but interrogating their origins and engaging with their effects to understand governance through ambivalence. Semi-legality can then be studied through migrants' lived experiences with regard to legal processes related to entry, stay and rights regimes (seeking out the 'mechanisms that allow them to be regular in one sense and irregular in another') as well as through state practices of 'uneven policy enforcement,' 'low repatriation rates,' 'mass regularization methods' and 'de facto tolerance of irregular presence' (Kubal, 2013). In Kubal's (2013: 555) own words, the concept of semi-legality helps to pinpoint the convenience of ambivalence for 'regimes, which claim that law and order are the main features distinguishing them from others' by pointing at the ways it allows states to balance fluid and fluctuating economic and political interests associated with alternating or simultaneous inclusion and exclusion. Accordingly, Kubal (2013: 555) sees semi-legality as 'not only tolerated,' but 'fueled and perpetuated' by states.

Nassar and Stel (2019) propose the concept of "strategic institutional ambiguity" to highlight that inconsistent policies directed at Syrians in Lebanon cannot be (fully) understood as failures. Conceptualizing ambivalence as institutional ambiguity, which can be seen in incomplete and vague policy formulations (as benchmarked vis-à-vis the experience of informed state and non-state, national and international policy experts), Nassar and Stel show that ambiguity in the realm of entry, stay and protection policies crucially depend on authorities pretending not to know things they demonstrably could have known, such as information publicly available or even presented to them, in order to legitimize policy inaction or vagueness. They thus suggest to engage with convenience through pinpointing how and why shifts between explicit and vague policy come about and exploring how such shifts align with three sets of consequences – marginalization of refugees, fragmentation of responsibilities, and securitization of engagement – that demonstrably

serve the interests of authorities to control refugee populations through minimal means. This leads them to conclude that institutional ambiguity is beneficial, and thus strategic, even if intentionality cannot be fully proven.

(8) *Convenient ambivalence in policy implementation*

Earlier, we discussed the idea of the "politics of uncertainty" to help make visible the convenience of *inaction* in policy-making on migration. Our final set of concepts on strategic non-regulation in migration governance also mobilizes the concept of uncertainty in a specific temporal sense to highlight the convenient *ambivalence* of policy implementation through migrants' experiences of uncertainty (rather than the experiences of street-level bureaucrats as in the above section) (Agier, 2008; Anderson, 2014; Brun, 2015; El-Shaarawi, 2015; Hasselberg, 2016; Franck, 2017; Hage, 2009; Khosravi, 2018). By definition, such studies are not invested into tackling the question of intent of state actors given that these are not their object of study. Grounded largely in anthropology or socio-legal studies, they delve both into European and non-European contexts to make sense of the "protracted uncertainty" (Biehl 2015; Horst and Grabska 2015), "politics of waiting" (Griffiths, 2013, 2014; Sanyal 2018), and "disorientation" (Tazzioli 2021, 2022). Working up from such migrant experiences, these works trace the functionality of ambivalence and look at the effects of implementation dynamics on migrants' everyday lives in terms of uncertainty, risk, opacity, liminality, informality. Of the incredible wealth of studies available on this, we highlight Griffith's (2013, 2014) conceptualization of 'governing through uncertainty' and Tazzioli's (2022) concept of 'disorientation.'

Griffiths (2013, 2014) applies extant literature on the 'politics of waiting' to cases of immigration detention in Britain. She proposes to study such temporal ambivalence through examining migrants' access to relevant information and tracing instances of confusion and miscommunication, specifically examining timeframes in relation to the unpredictable 'temporal ruptures,' or the alternation between endless waiting and sudden and dramatic change. In terms of operationalizing convenience, for Griffiths, the avoidable and hence willful aspects of the situations of her interlocutors are rendered researchable in the incongruity between extreme spatial control, confinement, and containment on the one hand, and radical temporal indeterminacy on the other hand. Imposed waiting and the stealing of time are understood as 'a technique of power, with governance through uncertainty constructing certain immigrants and extendible, transient and, ultimately, deportable' (Griffiths, 2013). The politics of waiting produces a passivity and desperation that undercuts migrants resistance to the system that seeks to expel them. Convenience is then implicitly understood as pinpointing the ways in which outcomes of specific policies and practices serve stated and unstated objectives of the authorities that have designed them. As with some other illustrations we have brought to bear, this reading of 'governing through uncertainty' (Griffiths, 2013) does not preclude conscious design, which Griffiths explicitly keeps on the table as an option, but analytically foregrounds convenience.

Looking at the experiences of asylum seekers in Greece, Tazzioli (2022) argues that "disorientation" as a manifestation of ambivalence should not be considered a side effect of refugee humanitarianism but, rather, a constitutive political technology of refugee governance. In line with many other scholars grappling with strategy that opt to analytically foreground convenience and the tracing of interests over the attempt to prove intent, Tazzioli (2022: 432) is

explicitly uninterested in the intentionality behind policies of disorientation. She argues that because "the boundaries between states' intentionality or non-intentionality are often quite blurred," it is more apt to focus on "disorientation that fragmented and disjointed knowledges generate on asylum seekers" rather than "what migration agencies know, don't know or disregard."

### *Synthesizing strategies to study strategic non-regulation*

As mentioned earlier, most of the studies discussed in the previous section largely 'stumbled' upon strategic non-regulation during the research process, rather than setting out to study it from the start. However, we believe that the phenomenon of strategic non-regulation should be more than an inductive by-product of migration research. It deserves to be an explicit subject of inquiry in its own right. Building on the above overview, this section therefore draws out some key focal questions for those interested in deliberately studying specific dimensions of strategic non-regulation in migration governance.

Reflecting on the previous section, both scholars engaging with inaction and researchers focusing om ambivalence showcase the importance of analytically foregrounding issues that are easy to overlook or dismiss. Studying strategic non-regulation, whatever form it takes, requires 'reading between the lines' and being attuned to glitches and apparent mistakes. They highlight the importance of revealing and interrogating a wide variety of gaps and inconsistencies in their data not as 'measurement errors' or 'thin data,' but as research findings in their own right that offer a relevant window onto the broader institutional context in which they are generated (Mazzei, 2003: 357). It is such silences and absences on the one hand and vaguenesses and ambiguities on the other that, the above overview suggest, scholars interested in strategic non-regulation would want to actively seek out in specific domains or scales of governance.

This always entails a critical juxtaposition of policy and practice, of statements and behavior, of what is said and what is known. This in turn, and as illustrated by the concepts leveraged above, requires a particular form of qualitative, triangulated, and contextual data and iterative, critical, and reflexive analysis that resonates with Olivier de Sardan's (2016: 121) 'anthropology of gaps, discrepancies and contradictions.' Following Stel (2020), whatever specific data is generated – from observations to interviews to documents – and whatever analytical strategy is used – from process tracing to archival analysis to institutional ethnography – , studying non-regulation always requires asking a specific set of questions: What is not being said? What is not being done? What is inconsistent? What is sensitive? And what is taken for granted?

When it comes to pinning down and understanding the potential strategy behind such non-regulation, the previous section shows that scholars working in this field use a variety of methods, scales and conceptual approaches. Those considering strategy predominantly as 'intent,' which are often concepts engaging with macro-level decision-making rooted in Law, Political Science, International Relations and Sociology, take a relatively top-down approach to seek out direct forms of 'evidence' to delineate their arguments. For them, confessions by state actors themselves are the ultimate standard of proof. While government officials sometimes adhere to official lines and may not always be forthcoming, direct acknowledgements of the unwillingness – rather than inability – to act or implement policy may be more frequent than one might think in

the course of immersive, qualitative research. Such stakeholder reflections on the respective salience of capacity and political will might be prompted in various ways.

When conducting interviews, researchers might confront interviewees with information and knowledge that is publicly available but is not used as a basis for policymaking. Inconsistencies between publicly known data and information versus what government officials are willing to acknowledge can reveal inconsistencies that are indicative of strategic non-regulation. Researchers could also perform an analysis of the inconsistencies between different legal instruments – laws, directives and regulations – and confront policymakers and bureaucrats with their findings, potentially leading to formal acknowledgements of strategic non-regulation (Frost forthcoming).

Researchers interested in intent in policymaking can also examine shifts from inaction to more resource-intensive action – whether this is a more inclusive policy, or a more repressive one – to retrospectively establish whether inaction was a choice or a capacity issue (Norman 2019, 2020; Mourad 2017). They might trace how local authorities navigate and diverge from central-level policies in order to demonstrate the awareness and intent of state officials in carrying out strategic non-regulation (Oomen et al. 2021; Stel, forthcoming). Finally, researchers might utilize process-tracing to understand why certain policy tools or decisions were mobilized or chosen by policymakers on a particular issue, and why potential alternatives were discarded (Mielke 2022).

Scholars approaching strategy as an accumulation of 'convenient' outcomes that can be studied via a more bottom-up tracing of interests draw on studies of local governance and micro- or meso-level implementation that can be primarily located in traditions of Anthropology and Political Geography. Rather than equating strategy with intent and deliberate or conscious action, they seek to study strategy through more indirect forms of 'evidence' that may 'shift the burden of proof' (i.e. 'if non-regulation serves your stated or unstated political and/or material interests it is on you to make credible this non-regulation was not deliberately designed or pursued').

This could entail turning to budgets as an indicator for institutional priorities and to identify areas of action versus inaction. For example, if a host country is heavily investing in border control versus contributing little to no investment in refugee care, researchers can investigate the discrepancy between the availability and utilization of state resources to pinpoint convenient inaction (Davies et al. 2017). It could also mean systematically comparing the differential treatment and policies between groups – such as migrants versus citizens, or different categories of migrant groups – to show that non-regulation does not follow from capacity or complexity issues but rather from political will.

What the reflexive synthesis above proposes is that no matter whether scholars are interested in evidencing intent or showcasing convenience, the empirical material collected should aim to demonstrate how strategic non-regulation is not a result of lacking capacity or institutional complexity, but rather how it serves state actors' interests by fulfilling one or all of its four functions: de facto outsourcing of service tasks, maximizing flexibility and consensus across different audiences, avoiding responsibility and accountability for decisions, and disciplining migrants and other stakeholders involved in migration governance.

## 5.    TOWARDS A NEW RESEARCH AGENDA ON MIGRATION GOVERNANCE

This paper sought to bring together emerging work on strategic non-regulation by highlighting its key dimensions and exploring the varying analytical approaches that scholars across a wide range of disciplines have used to capture and conceptualize it. It is important to reiterate that our paper is not an exhaustive overview of all studies on strategic non-regulation in migration governance. Rather, our contribution lies in gathering together and showcasing potential ways of approaching the study of strategic non-regulation and in analytically consolidating some of the commonalities as well as differences among recent works. By drawing out three primary dimensions – intent versus convenience, inaction versus ambivalence, and policymaking versus policy implementation – our aim was to facilitate the use and applicability of concepts to an even broader range of topics within migration governance.

In doing so, we hope to have also complicated three persistent binaries that continue to define the field of migration studies. First, our analysis contributes to a growing literature within migration studies that attempts to dismantle the epistemic divide between studies focusing on the so-called Global South versus North (Chimni, 1998; Garcés-Macareñas, 2018; Fiddian-Qasmiyeh, 2020). By including concepts derived from geographies that bridge this divide – especially the Middle East, North Africa and Europe – we call attention to how strategic non-regulation is used across states with varying 'capacities' and irrespective of broad forms of governance in place. While instances of strategic non-regulation originating from cases in the Global South are sometimes dismissed as simply the result of 'illiberal' or 'weak' governance characteristic of 'failed' or 'fragile' states, our analysis shows that allegedly 'strong' states in the Global North are just as likely to employ forms of strategic non-regulation (Stel, 2021).

Second, by putting in conversation conceptualizations of strategic non-regulation drawing on work with migrants categorized as refugees and on work with transnationally mobile people in a broader sense (Fiddian-Qasmiyeh et al., 2014), the paper signals the importance of exploring the continuities in the governance of migration rather than reproducing statist categorizations and exceptionalizations of mobility (Bakewell, 2008; Janmyr and Mourad, 2018; Zetter, 2007).

Third, in focusing on strategic non-regulation of state actors, the conceptualizations discussed in the paper tend to focus on the disciplinary effects of non-regulation: the ways in which policy gaps, discretionary implementation, and everyday unpredictability and uncertainty undermine migrants' collective subjectivities. Crucially, however, the literature on strategic non-regulation is in fact quite consistent in pointing out how non-regulation is not inherently bad for migrants and how people on the move also generate and appropriate forms of inaction and ambivalence. Tazzioli (2022) shows how asylum seekers to tactically utilize disorientation by state actors for their own purposes. Eule et al. (2019) elaborately discuss how the various strategies of coping and resistance of migrants and those that support them might make absent or contradictory regulations work for them. Norman (2020) points to how strategic non-regulation can provide flexibility and the opportunity for migrants and refugees to engage in de facto integration practices – through participation in informal economies, for example – that might not be permissible or

possible under a more rigidly controlled system of governance (Norman 2020). Stel (2016, 2020) shows how refugees can turn strategies of temporal liminality and spatial informalization against state authorities in aiming to secure residence and political protection. Thus, in bringing together the disciplinary strategies of state actors and the autonomy of migrants (Casas-Cortes et al., 2015; DeGenova, 2017; Tazzioli et al., 2018), the perspective of strategic non-regulation can connect different elements of the overarching governmentalities of migration.

Finally, the utility of studying strategic non-regulation as migration governance does not merely extend the field of migration studies but proposes crucial ways in which this field can inspire other disciplines and domains of study. Our analysis identified groupings of concepts that align with disciplinary divides and groupings that do not. For example, concepts originating within Law, Political Science, International Relations and Sociology tended to focus on the intent behind strategic non-regulation, whereas concepts drawn from Anthropology and Political Geography tend to focus on its convenience. Yet, the other two dimensions - a focus on inaction versus ambivalence and policymaking versus implementation – show that there are important commonalities across disciplines, indicating the potential for greater cross-fertilization and communication among scholars working within this field. Introducing strategic non-regulation as a focal point of migration scholarship, then, is a promising way to live up to the often-recognized but not always realized interdisciplinary potential of this field of study (Favell, 2022).

This, in turn, would enable migration scholars to not merely borrow *from* disciplinary literatures interested in what we have here discussed as strategic non-regulation – such as for instance work on discursive gaps in linguistics (Mazzei, 2003; Randazzo, 2015), hypocrisy in political science (Egnell, 2010; Krasner, 1999), and non-intervention in IR (Little, 1993) – but theoretically contribute *to* them. This is particularly the case for broader social science debates on policy failure (Castles, 2017; Chabal and Daloz, 1999; Ferguson, 1994). Our paper serves as a reminder that we can and should trace the political functionality of branding policies as failure and that in doing so political will is often a more pertinent focus of analysis than state capacity. In the end, migration governance is a weathervane for the workings of political power more broadly (Bakewell, 2014; Eule et al., 2018). Our synthesis of the ways in which strategic non-regulation is salient and can be studied in the domain of migration thus has ramifications beyond that domain, perhaps specifically for the study of the governance of society's 'marginalized' and 'undesirables' (Agier, 2008, Bayat, 1997, Chomsky, 2012 in Stel, 2020: 222) and the handling of all files 'sensitive' or 'classified' (Gould and Stel, 2021). Ultimately, policies and practices of rule do not emerge or prevail just despite of or in opposition to non-regulation, but operate through it (Stel, 2020: 223)

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
