# Peer review of "Strategic Non-Regulation as Migration Governance"

_Migration Politics_

## Round 1 · Referee Report · Barak Kalir · 2023-5-20

Report
I very much appreciate the attempt to offer a new umbrella term to capture the immense variety of strategic non-regulation (SNR) out there in the field of ‘migration management’ or mobility governance. This is a useful contribution for all the reasons mentioned in the article; and not least, because it will hopefully instigate a conversation and synergy between scholars and scholarship that, at the moment, might not recognize SNR as a common thread in their analysis.
The article is written in a fluent fashion and makes for a very engaging read. This is remarkable given the level of analytical complexity and the fact that it’s co-authored by three scholars. The balance that the paper strikes between working towards a high level of abstraction and staying close to empirical evidence is commendable.
I strongly recommend publishing the paper after the authors consider and attend to the following points (in no particular order):
- Not sure why showing that SNR is happening both in the Global North and the Global South helps ‘to dismantle the epistemic divide’. The ways in which we learn about the existence and operationalization of SNR in all these places is quite similar. How does it relate to epistemology? I see it more as an attempt to ‘de-exoticize’ the Global South or to indicate some entrenched (universal?) bureaucratic tendencies.
- On p. 10 you use the example of Colombian migrants (mostly refugees) in Ecuador to show how not all cases of non-regulation fall under SNR as an umbrella term. I read this as being slightly contradictory to the spirit of your suggested term. From my understanding, the term SNR captures a form of governance. That form can work in all directions; obviously, as you emphatically show, mostly favoring state actors, but possibly it can of course also be used by other actors and for different, even opposing, ends. In fact, this is precisely what you demonstrate with respect to the second binary in the conclusion.
- You might want to consider saying something about your positionality in conducting research on mobility in the MENA region as white, female, western, university educated scholars. What has your specific positionality led you to see? What might have it obscured? How did it condition your interlocutors to respond to your presence/questions/research aims in certain ways?
- You might want to consider that much of how bureaucracies work everywhere is the result of muddling through. Especially in the migration field, many policies are the outcome of a patchwork in which no one minister/ministry or even government can be seen as being the sole driver behind it. How should we think about SNR in view of the fact that policymaking is so often the end result of an accumulated pile of regulations, directives, amendments, etc. without a clear and coherent planning behind them. see for example: ‘The Deportation Mess: A Bureaucratic Muddling of State Fantasies’ (https://blogs.law.ox.ac.uk/research-subject-groups/centre-criminology/centreborder-criminologies/blog/2014/10/deportation-mess)
- Given that you are committed to show the full complexity and un/intended consequences of SNR, the reader is left a bit in a doubt whether – normatively speaking – your paper should call on us to preserve or do away with SNR (whenever we reveal its existence). Obviously, there is not a clear yes/no answer to this question (as the paper shows), but maybe you want to qualify your conclusion more clearly in this respect. Is our task as researchers to call out SNR irrespectively of the ways it’s put into work? Or should we also accompany it always with our normative judgement? Thinking here in the most pragmatic way, and playing the devil’s advocaat, are we not running a risk of playing into the hands of policymakers if we simply document and explicate the working of SNR? State actors can then finetune their ways of working around/with SNR to become even more aware of it and thus use it in even more efficiently repressive manners.

---

## Round 1 · Referee Report · Anonymous · 2023-9-5

Strengths
1. This article builds on a growing recognition that, ‘Scholars thus seem to agree that the strategic use of non-regulation by state actors is a significant aspect of migration governance.’ This article goes further, suggesting that non-regulation is a core feature of migration governance. Beyond this the article serves as a useful primer on a broader set of literature on non-reguilation from multiple disciplinary fields.
2. Valuable in helping to reduce the state-centrism and legalism in migration scholarship.
3. A generous engagement with a growing body of literature that has yet to adequately coalesce.
4. A potentially useful teaching tool.
Weaknesses
1. The conclusions are valuable but only hinted at early on. As such, the article comes across much more like a literature review than it needs to. The contributions are primarily synthetic and based on others’ work, but there is sufficient analytical content and originality to make more of this from the jump.
2. The introduction to the author’s scholarship is valuable in recognising their positionality and background. However, it now reads as somewhat decadent and perhaps a bit self-promotional. In a longer, book length piece this might be warranted but I found it somewhat distracting from the article’s central themes. Given that much of the author’s research (usefully) appears in the reviews, it also feels somewhat redundant.
3. My only substantive concern is that the authors are still remarkably state centric in their approach. They do well in conceptualising ‘governance’ as something done beyond states, but most of their discussion concentrates on the actions (or inactions) of national authorities and national level policy making (or lack thereof). There’s little about the private sector or national trade organisations and even less about local authorities who are (as we are increasingly being told) critical actors in the ‘localisation’ of migration governance. This includes policing, integration, housing, labour markets, etc. However, the words city, urban or municipal do not appear once in the piece. There is some discussion of local authorities, but largely in passing.
Report
This article seems like an invaluable resource that is truly in line with the journal's ambitions. With some relatively minor changes, I feel that it will be ready for publication.
Requested changes
1. Emphasise the analytical payoff and conclusions earlier and more robustly.
2. Revise the author introductions so as to better facilitate the narrative flow.
3. Speak more robustly about non-state and non-national actors' position in 'non-regulation'.

---

## Round 2 · Author Response

Thank you to the reviewers and editors for their helpful and thorough set of comments and suggested revisions. We discuss them in the next section ('List of Changes') according to comment and/or category. We truly appreciate the willingness of the reviewers to so thoughtfully engage with our paper. It is now stronger thanks to their attentiveness and consideration and we hope it will provide a valuable conceptual contribution for, as well as stimulate discussions amongst the readership of Migration Politics.

---

## Round 2 · List of Changes

Eliminating Section 2

Per the response of both the reviewers and editor, we choose to delete Section 2. We retained some of the vignettes and reflections in a new opening paragraph but deleted the separate section as suggested. We agree that this helps with the flow of the paper and focuses the attention on introducing and conceptualizing strategic non-regulation (SNR), rather than focusing on our previous collective work.

State Centrism

Both the editor and Reviewer 2 expressed concern about the paper being too state-centric. Although we fully acknowledge the relevance of SNR beyond the state (which is why we discuss migration governance and not government, as we explain in the respectively titled section), the choice to focus on state actors has been consciously made and explicitly argued in the article in a full paragraph on pgs. 5-6. It relates to both scope and demarcation (as the editor recognizes the extended state and non-state version of SNR would demand an additional paper) as well as content-related relevance: as we write, we are convinced that because “the very phenomenon of migration only exists due to the prevalence of an international nation-state system and its bordering practices,” an emphasis on state actors is merited. In response to the reviewer and editor observations, however, we have now put extra stress on the relevance of SNR for non-state actors and included this more explicitly in our future research agenda. We have also added some core references to this end.

Reviewer 2 also suggested including additional literature on actors within local governance. However, many of the concepts we discuss in the paper (especially those concerned with policy implementation) do engage with local authorities. These include: stand-offish policy-making, non-recording, the politics of non-knowledge, strategic ignorance, and disorientation. We therefore feel that our paper did already reference key works on local governance and have thus not included additional literature (also considering the already considerable length of the current bibliography).

The Global South/Global North Binary

Reviewer 1 was unsure why elucidating SNR would help to dismantle the epistemic divide between migration governance literature on the Global North and South. We partly argued that this divide is epistemic because its maps onto a disciplinary divide (Political Science, Public Administration and Sociology focus more on the Global North, while Geography and Anthropology are more likely to focus on the Global South), but we changed this in view of the reviewer’s comment as it raised questions. We have now elaborated throughout but specifically in the concluding section on how studying SNR impacts the divide, as far as we deemed appropriate within the scope of the current article (because this could indeed be a new paper too, as the editor remarks). In particular, we brought in the observation of how SNR allows us to de-exoticize the South and de-idealize the North, as suggested by a reviewer and the editor.

Removing Demarcating Examples

Reviewer 1 felt that including an example (previously on p. 10, the case of Colombian migrants (mostly refugees) in Ecuador) to show how not all cases of non-regulation fall under SNR as an umbrella term undermines the aim of the article. We have had a lot of internal discussion while writing the article on the helpfulness of introducing such ‘boundary’ cases, and the observation by the reviewer helped us to decide that such examples add rather than alleviate confusion. We therefore decided to eliminate the paragraph in question.

Positionality

Reviewer 1 suggested adding comments about our positionality as researchers (white, female, western, university educated scholars). Since the paper is fundamentally a theoretical rather than an empirical one (and especially since we chose to remove the original section 2, as per other reviewer comments), we opted not to devote precious space to our own positionality. We did, however, add more general reflections on the role of positionality in encountering, experiencing and understanding SNR in our section on synthesizing empirical strategies to study non-regulation. We thank the reviewer for having raised the issue of positionality, as we did not ultimately plan to address it but in fact consider it to be a crucial element for our methodological discussion towards the end of the paper.

Bureaucracies and ‘Muddling Through’

Reviewer 1 advocated for including a discussion of how bureaucracies work everywhere and the concept of ‘muddling through.’ We very much agree that this is a crucial element of migration governance worldwide. However, for us, the notion of muddling through adds to explanations for inaction and ambivalence that are not strategic (such as compromise, capacity, and complexity) and that – while relevant – do not add to the focus of our paper that sets out to explore elements of non-regulation that are strategic. To clarify this, we have now included a discussion of this concept and the relevant reference in the paper at two instances (in the introduction paragraph and in the section reflecting on ‘strategy’).

Normative Implications of SNR

Reviewer 1 asked whether explicating SNR could play into the hands of policymakers that want to further exclude or repress migrants. We very much welcome this reflection, which indeed could be a complete political reflexivity paper in its own right. We have added further reflection to this end in a new paragraph in the concluding section.

Analytical Implications

Reviewer 2 proposed emphasizing the analytical conclusions earlier on in the paper. We have rephrased the paragraph outlining contributions in the abstract to be both more extensive and more assertive. Also, we feel that by doing away with section 2, the paper moves towards core debates and conclusions earlier on.

---

## Editorial Decision

unknown